# The Association of Leptin with Left Ventricular Hypertrophy in End-Stage Kidney Disease Patients on Dialysis

**DOI:** 10.3390/biomedicines11041026

**Published:** 2023-03-27

**Authors:** Susana Coimbra, Cristina Catarino, Maria Sameiro Faria, José Pedro L. Nunes, Susana Rocha, Maria João Valente, Petronila Rocha-Pereira, Elsa Bronze-da-Rocha, Nuno Bettencourt, Ana Beco, Sofia Homem de Melo Marques, José Gerardo Oliveira, José Madureira, João Carlos Fernandes, Vasco Miranda, Luís Belo, Alice Santos-Silva

**Affiliations:** 1UCIBIO—Applied Molecular Biosciences Unit, Department of Biological Sciences, Faculdade de Farmácia, Universidade do Porto, 4050-313 Porto, Portugal; 2Associate Laboratory i4HB, Institute for Health and Bioeconomy, Faculdade de Farmácia, Universidade do Porto, 4050-313 Porto, Portugal; 3TOXRUN—Toxicology Research Unit, University Institute of Health Sciences, CESPU, CRL, 4585-116 Gandra, Portugal; 4Hemodialysis Clinic Hospital Agostinho Ribeiro, 4610-106 Felgueiras, Portugal; 5Faculty of Medicine, University of Porto, 4200-319 Porto, Portugal; 6National Food Institute, Technical University of Denmark, 2800 Kongens Lyngby, Denmark; 7Health Science Research Centre, University of Beira Interior, 6201-001 Covilhã, Portugal; 8Hemodialysis Clinic of Porto (CHP), 4200-227 Porto, Portugal; 9Center for Health Technology and Services Research (CINTESIS), Faculty of Medicine, University of Porto, 4200-450 Porto, Portugal; 10NefroServe, Hemodialysis Clinic of Barcelos, 4750-110 Barcelos, Portugal; 11NefroServe Hemodialysis Clinic of Viana do Castelo, 4900-281 Viana do Castelo, Portugal; 12Hemodialysis Clinic of Gondomar, 4420-086 Gondomar, Portugal

**Keywords:** leptin, left ventricular mass index, NT-proBNP, hemoglobin, calcium, GDF-15

## Abstract

Left ventricular hypertrophy (LVH) is a common cardiovascular complication in end-stage kidney disease (ESKD) patients. We aimed at studying the association of LVH with adiponectin and leptin levels, cardiovascular stress/injury biomarkers and nutritional status in these patients. We evaluated the LV mass (LVM) and calculated the LVM index (LVMI) in 196 ESKD patients on dialysis; the levels of hemoglobin, calcium, phosphorus, parathyroid hormone, albumin, adiponectin, leptin, N-terminal pro B-type natriuretic peptide (NT-proBNP) and growth differentiation factor (GDF)-15 were analyzed. ESKD patients with LVH (*n* = 131) presented higher NT-proBNP and GDF-15, lower hemoglobin and, after adjustment for gender, lower leptin levels compared with non-LVH patients. LVH females also showed lower leptin than the non-LVH female group. In the LVH group, LVMI presented a negative correlation with leptin and a positive correlation with NT-proBNP. Leptin emerged as an independent determinant of LVMI in both groups, and NT-proBNP in the LVH group. Low hemoglobin and leptin and increased calcium, NT-proBNP and dialysis vintage are associated with an increased risk of developing LVH. In ESKD patients on dialysis, LVH is associated with lower leptin values (especially in women), which are negatively correlated with LVMI, and with higher levels of biomarkers of myocardial stress/injury. Leptin and NT-proBNP appear as independent determinants of LVMI; dialysis vintage, hemoglobin, calcium, NT-proBNP and leptin emerged as predicting markers for LVH development. Further studies are needed to better understand the role of leptin in LVH in ESKD patients.

## 1. Introduction

Cardiovascular diseases (CVD), such as coronary artery disease, congestive heart failure, arrhythmias and sudden death, are the major causes of mortality and morbidity in chronic kidney disease (CKD), especially in end-stage kidney disease (ESKD) patients. Left ventricular hypertrophy (LVH) is a common cardiovascular (CV) complication in CKD, with a negative prognostic value, contributing to diastolic dysfunction, congestive heart failure, arrhythmia and sudden death [1,2]; in fact, it is considered a risk predictor for CV morbidity and mortality among CKD patients [2]. In the intermediate stages of CKD, LVH may affect 50–70% of patients, and up to 90% of ESKD patients on dialysis [3]. The development of LVH in CKD seems to involve several factors, leading to myocardial cell thickening and concentric LV remodeling, and is often associated with activation of the renin-angiotensin system [4].

N-terminal pro B-type natriuretic peptide (NT-proBNP) is synthesized within cardiac myocytes in response to cardiac wall stress, being a sensitive marker of ventricular dysfunction. It has been widely used in the diagnosis and monitoring of heart failure; however, its diagnostic value appears to be reduced with renal function loss [5].

Growth differentiation factor (GDF)-15 belongs to the transforming growth factor-β family and it is expressed in cardiomyocytes, kidney tubular cells, adipocytes, macrophages, endothelial cells and vascular smooth muscle cells [6]. Increased blood levels of GDF-15 have been reported in tissue injuries and inflammatory states, both associated with cardiometabolic risk. Moreover, higher levels of GDF-15 have been shown to correlate with heart failure and with increased mortality in CKD [7]; in fact, GDF-15 has been proposed as a helpful tool for the diagnosis of heart failure in dialysis patients [8]. 

Furthermore, an increase in both GDF-15 and NT-proBNP has been linked to a higher risk for progression of CKD [6].

ESKD patients on dialysis may develop disturbances in nutritional status, evidenced by clinical nutritional biomarkers, such as body weight and composition, and by biological nutritional biomarkers, such as serum albumin levels [9]. Malnutrition in ESKD patients on dialysis treatment has been associated with increased morbidity and mortality, mainly due to the coexistence of CV and infectious complications in these patients [9].

The adipokines adiponectin and leptin are known as important mediators of cardiometabolic risk in obesity; however, their role in ESKD patients is poorly clarified.

Adiponectin, mainly synthesized by adipocytes, has anti-inflammatory, insulin-sensitizing and anti-atherogenic properties. Its levels are enhanced in ESKD patients and seem to present a negative correlation with body mass index (BMI) in these patients [10,11]. The higher adiponectin levels in ESKD might favor atheroprotective modifications in high-density lipoprotein subfractions, and also protect low-density lipoprotein particles from oxidative atherogenic changes [12]. A study conducted in ESKD patients on dialysis with type 2 diabetes mellitus showed that higher adiponectin levels were associated with LVH, and that the adiponectin levels could be modulated by a chronic hypervolemic state in these patients [13]. Hyperleptinemia has been associated with inflammation, insulin resistance, protein energy wasting and progression of CKD [14]. The increased leptin levels reported in CKD patients [15] were pointed at as promoters of endothelial dysfunction, which is crucial for atherosclerosis development [16]. In peritoneal dialysis patients, leptin levels above 40 ng/mL were associated with LVH [17]. However, a study in ESKD patients under hemodialysis showed low leptin values as an independent predictor of mortality [18]. Decreased leptin levels were also associated with a higher risk of CV events in hemodialysis patients and was also considered an independent risk factor for LVH development [19]. Interestingly, in ESKD, while patients with a history of stroke presented higher leptin levels than those without a stroke history, patients with congestive heart failure showed lower leptin than those without a history of congestive heart failure [20]. Given the controversial data, further studies are warranted to fully clarify leptin’s role, namely, its association with morbidity and mortality risks and its contribution to LVH development in CKD and, especially, in ESKD.

We aimed at studying the association of LVH with adipokine levels (adiponectin and leptin), CV stress/injury (NT-proBNP and GDF-15) and nutritional status (BMI and albumin) in ESKD patients on dialysis. 

## 2. Materials and Methods

### 2.1. Patients

This study was conducted in collaboration with 5 Dialysis Clinics from the North of Portugal (from 2016–2019), after approval by the Committee on Ethics from Faculty of Pharmacy, Porto University, and by the Directors of Dialysis Clinics (Report No. 26 April 2016). Patients signed a written informed consent to participate in the study, and their privacy rights were respected; subjects presenting autoimmune diseases, malignancy or active infectious diseases were excluded from the study.

The study included 196 ESKD patients under dialysis therapy for at least 90 days. Dialysis was performed using FX-class^®^ high-flux polysulfone dialyzers (Fresenius Medical Care, Bad Homburg, Germany); 27 patients were under high-flux hemodialysis, while 169 were under on-line hemodiafiltration. Dialysis was performed 3 times/week, for 3–5 h in each session; patients were on dialysis treatment for a median period of 3.24 (1.60–6.37) years. Dialysis clearance of urea was expressed as Kt/V and eKt/V. CKD etiologies included diabetic nephropathy (74 patients), hypertensive nephrosclerosis (19 patients), chronic glomerulonephritis (14 patients), polycystic kidney disease (11 patients) and other diseases or uncertain etiology (78 patients). Concerning the vascular access for the dialysis procedure, 28 patients used a central venous catheter, 160 used an arteriovenous fistula and 8 patients used an arteriovenous graft.

Echocardiography is the main tool used to evaluate the LV mass (LVM) in clinical practice [4]; accordingly, we evaluated the LVM through echocardiographic studies, correcting for body surface area, as recommended, and presented the values as the LVM index (LVMI). In accordance with the American and European guidelines [21], LVH was defined by an LVMI > 115 g/m^2^ in men, and an LVMI > 95 g/m^2^ in women. The patients were divided into two groups: LVH (*n* = 131) and non-LVH (*n* = 65).

### 2.2. Analytical Assays

Blood was collected into tubes without an anticoagulant and with ethylenediaminetetraacetic acid immediately before a midweek dialysis session to obtain serum, whole blood and plasma. The samples were processed within 2 h of collection; aliquots of plasma and serum were prepared and stored at −80 °C until assayed.

The hemoglobin was evaluated by an automatic blood cell counter (Sysmex K1000; Sysmex, Hamburg, Germany). The serum levels of albumin, calcium and phosphorus were measured by routine procedures, using an auto-analyser (Cobas Integra 400 Plus, Roche Diagnostics, Basel, Switzerland). The serum concentrations of parathyroid hormone (PTH) were determined by chemiluminescence immunoassay (Advia Centaur XP, Siemens, Erlangen, Germany).

The circulating levels of adiponectin, leptin, NT-proBNP and GDF-15 were analyzed through commercial enzyme-linked immunosorbent assay (ELISA) kits (human total adiponectin and human leptin, R&D Systems Inc., Minneapolis, MN, USA; proBNP human ELISA kit and human GDF-15 ELISA Kit, Abcam, Cambridge, UK, respectively). 

### 2.3. Statistical Analysis

The Statistical Package for Social Sciences (SPSS, v.28.0, Chicago, IL, USA) for Windows was used. The data distribution was evaluated by Kolmogorov–Smirnov analysis. The results are presented as mean ± standard deviation (SD) or as median (interquartile range) for variables with normal or non-normal distribution, respectively. For comparison between the groups, we used, for continuous variables, the Mann–Whitney U test and the unpaired Student *t*-test, in accordance with the Gaussian distribution of the variables; for categorical variables, a chi-squared test and Fisher’s exact test were employed. The strength of the correlations between the variables was determined through Spearman’s rank correlation coefficient. To evaluate the contribution of different variables (leptin, adiponectin, NT-proBNP, GDF-15, calcium, hemoglobin, urea reduction ratio (URR) and eKt/V) to the LVMI values, a multiple regression analysis was performed using stepwise selection, with an entry criterion of *p* < 0.05; the data presenting a non-Gaussian distribution were transformed to data with a normal distribution using the Templeton method [22]. To determine the effects of age, gender, BMI, dialysis vintage and dialysis parameters (URR, Kt/V, eKt/V, ultrafiltration volume), systolic and diastolic blood pressure, and levels of calcium, phosphorous, PTH, albumin, hemoglobin, NT-proBNP, GDF-15, leptin and adiponectin on the likelihood of developing LVH, a binary logistic regression was conducted. A *p* value lower than 0.05 was considered as statistically significant.

## 3. Results

The data for the ESKD patients with and without LVH is presented in Table 1; the prevalence of LVH was 66.8%. The two groups of patients were matched for age, gender, BMI, dialysis vintage, dialysis adequacy (URR, Kt/V, eKt/V) and systolic and diastolic blood pressures. ESKD patients with LVH presented significantly higher levels of NT-proBNP, GDF-15 and calcium, and significantly lower levels of hemoglobin when compared with non-LVH patients; a trend towards lower leptin values was also observed (*p* = 0.052; *p* < 0.001 after adjustment for gender as a confounding factor). No significant differences were observed between the groups for albumin, phosphorus, PTH and adiponectin.

Since leptin levels differ between males and females, and its relationship with LVH seems to vary with gender [23], we evaluated and compared leptin concentrations presented by males and females from the two groups (Figure 1). ESKD females with LVH (33.5 (7.9–70.9) ng/mL) showed significantly lower (*p* = 0.045) leptin values than ESKD females without LVH (48.8 (20.8–138.1) ng/mL); males with LVH (6.8 (2.3–15.0) ng/mL) showed a trend towards lower values (*p* = 0.118) when compared to males without LVH (8.9 (4.9–25.7) ng/mL).

As depicted in Figure 2, ESKD males with LVH (11.3 (10.7–12.1) g/dL) presented significantly lower hemoglobin values (*p* = 0.029) than ESKD males without LVH (12.0 (11.3–12.7) g/dL); no significant differences (*p* = 0.515) were found between ESKD females with (11.3 (10.5–11.8) g/dL) or without LVH (11.4 (10.9–11.9) g/dL).

As observed in Figure 3, ESKD males with LVH (17.9 (10.0–30.9) g/dL) presented significantly higher NT-proBNP values (*p* = 0.003) than ESKD males without LVH (10.9 (5.7–17.3) g/dL); no significant differences (*p* = 0.188) were found between ESKD females with (13.6 (8.0–25.8) g/dL) or without LVH (10.2 (7.2–23.4) g/dL). For calcium and GDF-15 levels, no significant differences were observed between females with or without LVH or between males with or without LVH.

Considering the association of adipokine levels with BMI, we divided both groups—LVH and non-LVH ESKD patients—in two subgroups according to their BMI, with the threshold set at 25 kg/m^2^. In both the LVH and non-LVH groups, the leptin levels are higher (*p* < 0.001, in both cases) in patients with a BMI > 25 kg/m^2^ (23.2 (8.1–70.7) ng/mL, *n* = 71 and 30.9 (17.5–107.9) ng/mL, *n* = 38, respectively), as compared with individuals with a lower BMI (5.5 (2.1–20.1) ng/mL, *n* = 60 and 6.7 (2.8–23.6) ng/mL, *n* = 27, respectively). 

We found that the leptin levels in LVH patients showed a trend towards lower values when compared to non-LVH patients. The adipose secretion of leptin is known to be higher in women [23]. Therefore, we performed an additional adjustment for gender, detecting a significant increase in leptin values for the LVH group, including both genders. Considering the BMI value, it is worth noting that leptin values increased with increasing BMI in both groups. 

Concerning adiponectin, both the LVH and non-LVH groups showed lower values (*p* < 0.001 and *p* = 0.018, respectively) for patients with a high BMI (10.0 (5.4–15.1) ng/mL, *n* = 71 and 10.2 (6.4–13.1) ng/mL, *n* = 38, respectively) than for patients with a BMI < 25 kg/m^2^ (17.9 (10.2–24.1) ng/mL, *n* = 60 and 16.1 (9.4–19.1) ng/mL, *n* = 27, respectively).

In the LVH group, the LVMI presented significant positive correlations with NT-proBNP (*r_S_* = 0.308, *p* < 0.001) and with adiponectin (*r_S_* = 0.242, *p* = 0.005), and significant negative correlations with leptin (*r_S_*= −0.378, *p* < 0.001; *r_S_* = −0.311, *p* = 0.009 and *r_S_* = −0.252, *p* = 0.048, for men and women, respectively) and with the markers of dialysis adequacy, URR (*r_S_* = −0.182, *p* = 0.037) and eKt/V (*r_S_* = −0.181, *p* = 0.038). Moreover, in the LVH group, leptin correlated negatively with NT-proBNP (*r_S_* = −0.322; *p* < 0.001), both in men (*r_S_* = −0.277, *p* = 0.021) and women (*r_S_* = −0.287, *p* = 0.024). 

In ESKD patients without LVH, the LVMI was negatively correlated with leptin (*r_S_* = −0.257, *p* = 0.039); however, when analyzing the data according to gender, the correlation was lost (*r_S_* = 0.176, *p* = 0.312 and *r_S_* = −0.068, *p* = 0.719, for men and women, respectively).

To evaluate the strength of the correlations between LVMI and the studied variables, we performed a multiple linear regression analysis for all the patients; of the tested variables, leptin and NT-proBNP were significantly related with LVMI (Table 2).

As already mentioned, a binary logistic regression was also conducted to determine the effects of all the variables studied (clinical and analytical) on the risk of developing LVH. When entering all the parameters into the regression model, decreasing hemoglobin and increasing calcium and NT-proBNP levels, as well as dialysis vintage, were significantly associated with an increased likelihood of developing LVH in a statistically significant model (χ^2^ of 40.256, *p* = 0.003) that explained 25.8% of the variance in LVH prevalence, and correctly classified 71.9% of cases. When using a forward stepwise method for adding independent variables into the regression, we obtained a statistically significant model (χ^2^ of 19.017, *p* < 0.001) encompassing the decreasing hemoglobin and leptin levels in addition to the increasing calcium and dialysis vintage, as the predicting parameters of LVH. This model correctly classified 68.9% of LVH cases, though it could only explain 12.9% of the variance in LVH prevalence.

## 4. Discussion

The association of leptin levels with CV outcomes in ESKD patients on dialysis treatment is controversial. According to our data, lower leptin values in ESKD patients on dialysis (especially in women) is associated with the occurrence of LVH, which is negatively correlated with LVMI. Additionally, lower leptin values appear to be an independent determinant of LVMI and also appear to be associated with an increased risk of developing LVH. These findings may seem controversial when considering that increasing leptin levels have been associated with CVD. Despite this, our data are in agreement with other studies focused on LVH. A study by Allison et al. [24] of 1464 multi-ethnic participants reported a significant negative association of leptin with LV structure and function, and with LVH. Other studies of 410 Spanish adults [25], 432 participants of the community-based Framingham Heart Study [26] and 39 sedentary postmenopausal women [27] have also reported a negative association of leptin with LVM and LV thickness. Another study found that higher leptin levels are associated with lower LVM and stiffness in obese black women [23]. Leptin concentrations were also inversely correlated with LVMI in hypertensive patients with a BMI < 25 kg/m^2^ [28]. Recently, it was reported that decreased leptin levels were an independent risk factor for LVH development in 165 patients on hemodialysis [19]. Actually, as far as we know, the latter study is the only one performed in hemodialysis patients reporting the association between low leptin concentrations and LVH.

A study performed in normal weight, overweight and early obese subjects suggested that higher leptin levels could contribute to reducing LVM by inhibiting the deposition of triglycerides in the myocardium [24]. In addition, by increasing myocardial AMP-activated protein kinase phosphorylation, leptin may contribute to enhance fatty acid oxidation, favoring the decrease of lipid stores [29]. The inverse association of leptin with LVH is reinforced by studies performed in mice, reporting the development of LVH in the animals lacking leptin or its receptor; moreover, it was found that hypertrophy regressed when the mice received leptin afterward [30]. Administration of adenoviral leptin also improved cardiomyopathy and myocardial steatosis in transgenic mice [29]. Leptin treatment in ob/ob mice resulted in an increase of catalytic activity of phosphoinositide 3-kinase and a reduction in myocyte apoptosis and in the development of LVH [31]. Thus, lower leptin values seem to predispose patients to an increased LVH risk, while increased levels appear to have a protective effect on the development of LVH. 

Recently, it was reported that leptin circulating levels can independently predict CV outcome and all-cause mortality in hemodialysis patients, and this may result from leptin effects on the development of LVH and peripheral vascular disease [19]. According to Qin et al. [19], changes in leptin signaling pathways may induce disturbances in the metabolism of glucose and of other substrates, leading to a poor CV outcome. Another possible cause for the association of leptin with a poor outcome in hemodialysis patients is malnutrition [19]. In our study, although we did not find significant differences in albumin and BMI levels between the LVH and non-LVH groups, we cannot exclude this hypothesis, which should be further clarified by using more sensitive nutritional biomarkers.

Kamimura et al. reported that leptin may have a higher impact on the LV structure and function in women [23]. Actually, when compared to men, women have different fat storage patterns, present enhanced cardiac work and lesser myocardial efficiency [32]; women also show increased fatty acid utilization in LVH conditions [33]. Thus, it is reasonable to hypothesize that women are more prone to cardiac injuries and that decreased leptin values may contribute to this toxic effect. In accordance, in the present study, we found that ESKD women presenting LVH showed significantly lower levels of leptin. 

In the study by Kamimura et al. [23], obese women presenting higher leptin levels showed less LV mass and stiffness, as compared to lean women, raising the hypothesis that in obese patients with resistance to the weight-reduction leptin action, the higher leptin levels exert a direct effect on preventing the development of LVH [34]. In line with this hypothesis, in our study, including patients with and without LVH, matched for BMI, none of the patients presenting LVH were obese (median BMI of 25.6 kg/m^2^). The studies by Pladevall et al. [25], Lieb et al. [26] and Di Blasio et al. [27] including normal weight individuals also reported the same relationship between leptin and LVH. Independently of weight, the impact of lower leptin levels in lipid metabolism may explain the reported data.

We found an inverse correlation of LVMI with leptin and a positive correlation with NT-proBNP; by performing a multiple linear regression analysis, leptin and NT-proBNP emerged as independent determinants of LVMI for the LVH group. Moreover, NT-proBNP, leptin, calcium and hemoglobin showed potential as predicting biomarkers of LVH. 

The well-known biomarker of myocardial stress, NT-proBNP, was shown to be higher in the LVH group than in the non-LVH group of ESKD patients, especially in males; a positive correlation between this biomarker and LVMI was also found for the LVH group of patients. Supporting the inverse association observed between leptin and LVH, we found a negative correlation between leptin and NT-proBNP. Moreover, an increased level of NT-proBNP appears as a predicting marker of LVH development. An increase of stress on the left atrial and ventricular walls induces an increase in NT-proBNP, thus, the relation between increased NT-proBNP levels, high LVMI and LVH development comes as no surprise. 

Also documented in the literature [35] is chronic anemia as a common condition in ESKD patients, which is usually treated with erythropoietic stimulating agents and/or iron supplementation, according to current guidelines [36]. In response to lower levels of hemoglobin, the adaptive mechanisms to hypoxia are triggered, namely an increase in cardiac output and blood flow [37], favoring LVH development [38]. A high cardiac output state may trigger the remodeling of the LV, favoring the dilation of the ventricle (in response to chronic volume overload) and thickening of the wall, in an attempt to decrease the wall tension of the dilated ventricle. In accordance, we found that LVH patients, especially ESKD males, presented lower hemoglobin levels than the non-LVH group. Moreover, in the binary logistic regression, in both methods used, low hemoglobin emerged as a predictor of LVH development. 

In CKD, disturbances in calcium-phosphorus metabolism may have a negative impact in cardiac structure and function [39]. A positive calcium balance may increase the risk of vascular calcification [40], which is known to enhance the risk of LVH [41]. Patients with diabetes and LVH were found to present significantly higher calcium levels than those with diabetes but without LVH [42]. In line with this, we found higher levels of calcium in ESKD patients with LVH, which were also shown to be associated with a higher risk of developing LVH.

ESKD is a wasting illness, and it is known that patients undergoing dialysis present a higher risk of developing comorbidities, such as CVD. Although dialysis vintage has been associated with an increased mortality risk in ESKD patients, the specific cause of mortality is not clear [43]. Indeed, it was reported that dialysis vintage may contribute more to a higher risk for infection-related mortality than CVD mortality [43]. In our study, long-term dialysis is pointed at as a predictor of developing LVH in ESKD patients, further contributing to the increased risk for CVD events.

LVMI correlated inversely with markers of dialysis adequacy, URR and eKt/V, suggesting that a decrease in the removal of waste products creates a favorable milieu for LVH. 

Considering the other adipokine, adiponectin, we found a positive correlation with LVMI. It has been reported that the relationship between adiponectin and LVMI varies with the risk of LVH, showing a negative correlation in subjects at low risk of LVH, and a positive correlation in subjects at high risk of LVH [44]. Higher adiponectin was also associated with LVH in ESKD diabetic patients on dialysis [13]. Considering LVH and non-LVH groups, no significant differences between adiponectin levels were found. The multiple linear regression analysis did not show adiponectin as an independent determinant for LVMI, and in the binary logistic regression analysis, adiponectin was not considered a predicting marker of LVH. Our data suggest a weak association of this adipokine with the LV structure and function in ESKD patients on dialysis.

Concerning GDF-15, we found a higher plasma concentration in ESKD patients with LVH than in the non-LVH group. In fact, it has been reported that cardiac biomarkers, such as GDF-15 and NT-proBNP, are significantly associated with LVH [45]; however, we did not find a correlation between LVMI and GDF-15 levels in ESKD patients on dialysis, as it was reported for hypertensive patients, as compared to healthy individuals [46,47]. The role of GDF-15 in LVH development is still controversial. A study in mice showed that GDF-15 antagonizes the hypertrophic response and loss of ventricular performance, possibly through a mechanism involving SMAD proteins [48]. Another study showed that GDF-15 blocks norepinephrine-induced myocardial hypertrophy through a pathway involving inhibition of the epidermal growth factor receptor transactivation [49]. The induction of GDF-15 in the heart, as a defense mechanism to protect from ischemia/reperfusion injury, was also shown [50]. It seems that an increase in GDF-15 is involved in protective mechanisms to counteract hypertrophy and is not a determinant for LVH development.

The complexity of comorbid conditions and multidrug therapy (e.g., statins, diuretics, antihypertensives, erythropoiesis stimulating agents, iron) represents a limitation when interpreting data from ESKD patients; studies using larger samples would certainly reinforce the robustness of the data, as well the performance of longitudinal studies. Nonetheless, in the present work, we performed a multiple regression analysis, allowing us to study the contribution of different variables to LVMI values; we also performed a binary logistic regression analysis, allowing us to determine the impact of each studied parameter on the likelihood of developing LVH, contributing to the establishment of more accurate associations of LVH with the studied biomarkers and to reinforce the conclusions.

A summary of our main findings concerning the predictive biomarkers of LVH in ESKD is presented in Figure 4.

## 5. Conclusions

In conclusion, our data demonstrated that, in ESKD patients on dialysis, LVH is associated with higher myocardial stress/injury, as shown by the higher levels of NT-proBNP and GDF-15; lower leptin levels that are negatively correlated with LVMI; leptin and NT-proBNP appeared as independent predictors of LVMI; and, finally, the association of reduced levels of hemoglobin and leptin, and increased dialysis vintage, calcium and NT-proBNP with an increased likelihood of developing LVH. Our data suggest that further studies are needed to better understand the role of leptin in LVH in ESKD patients on dialysis treatment, since lower leptin values that are commonly assumed to be protective for CVD in the general population seem to predispose these patients to an increased LVH risk, and increased leptin values appear to have a protective effect on the development of LVH in these patients.

## Figures and Tables

**Figure 1 biomedicines-11-01026-f001:**
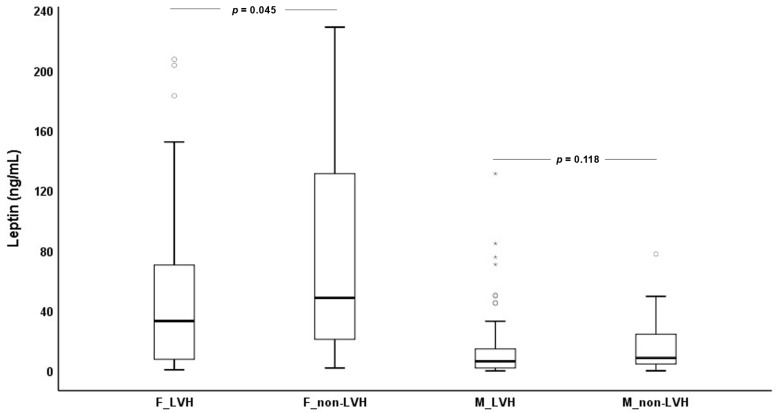
Leptin values for females (F) and males (M) with (LVH) and without left ventricular hypertrophy (non-LVH).

**Figure 2 biomedicines-11-01026-f002:**
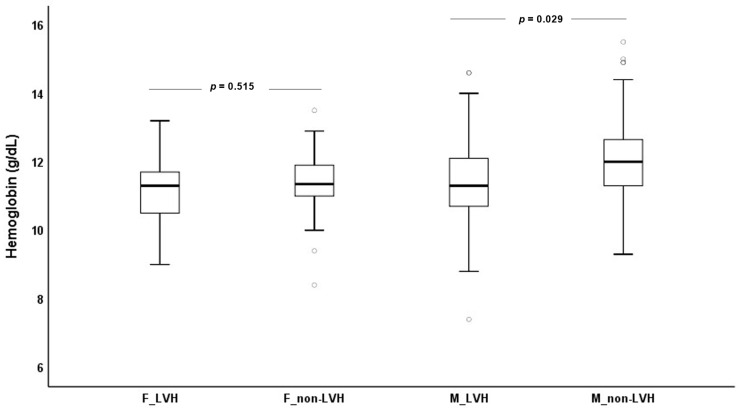
Hemoglobin concentrations for females (F) and males (M) with (LVH) and without left ventricular hypertrophy (non-LVH).

**Figure 3 biomedicines-11-01026-f003:**
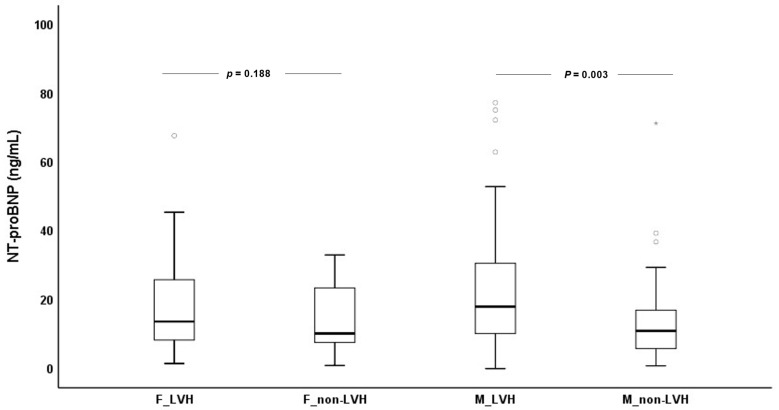
N-terminal pro B-type natriuretic peptide (NT-proBNP) concentrations for females (F) and males (M) with (LVH) and without left ventricular hypertrophy (non-LVH).

**Figure 4 biomedicines-11-01026-f004:**
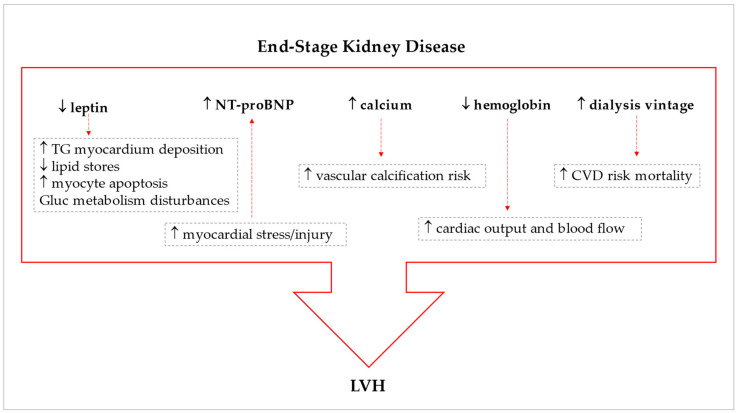
Schematic view of the predicting biomarkers of left ventricular hypertrophy and potential underlying mechanisms in end-stage kidney disease patients on dialysis, the red arrows indicating potential mechanistic consequences. CVD, cardiovascular disease; Gluc, glucose; LVH, left ventricular hypertrophy; NT-proBNP, N-terminal pro B-type natriuretic peptide; TG, triglycerides. ↑, higher/increased; ↓ lower/decreased.

**Table 1 biomedicines-11-01026-t001:** Demographic, dialysis-related and biochemical data for end-stage renal disease patients without and with left ventricular hypertrophy.

	non-LVH (*n* = 65)	LVH (*n* = 131)	*p* Value
Age (years)	70.2 (60.5–79.5)	71.8 (60.7–79.5)	0.707
Gender (*n*, Male/Female)	35/30	69/62	0.878
BMI (kg/m^2^)	26.7 ± 4.6	25.6 ± 4.3	0.129
Dialysis Vintage (years)	2.5 (1.4–5.4)	3.6 (1.8–7.1)	0.081
URR (%)	80.0 (76.0–83.0)	79.0 (75.0–83.0)	0.418
Kt/V	1.82 ± 0.31	1.77 ± 0.32	0.349
eKt/V	1.66 ± 0.34	1.60 ± 0.30	0.218
Ultrafiltration Volume (L)	2.46 ± 0.88	2.37 ± 0.96	0.523
SBP (mmHg)	137 (126–148)	137 (126–156)	0.313
DBP (mmHg)	63 (56–72)	61 (54–72)	0.512
Calcium (mg/dL)	8.93 ± 0.50	9.11 ± 0.55	0.028
Phosphorus (mg/dL)	4.20 (3.51–5.04)	4.17 (3.30–4.80)	0.307
PTH (pg/dL)	349 (208–514)	354 (212–510)	0.622
Albumin (g/dL)	4.00 (3.75–4.10)	3.90 (3.60–4.20)	0.170
Hemoglobin (g/dL)	11.5 (11.0–12.5)	11.3 (10.6–12.0)	0.039
NT-proBNP (ng/mL)	10.8 (6.2–17.6)	16.5 (9.0–29.8)	0.002
GDF-15 (ng/mL)	10.6 (7.8–12.8)	11.3 (8.4–14.5)	0.048
Leptin (ng/mL)	21.3 (6.7–51.2)	13.7 (4.6–42.7)	0.052 *
Adiponectin (ng/mL)	10.8 (7.9–17.6)	12.2 (7.2–20.1)	0.503

* *p* < 0.001 after adjustment for gender as a confounding factor. The data are presented as mean ± standard deviation or as median (interquartile range). BMI, body mass index; DBP, diastolic blood pressure; GDF-15, growth differentiation factor; LVH, left ventricular hypertrophy; NT-proBNP, N-terminal pro B-type natriuretic peptide; PTH, parathyroid hormone; SBP, systolic blood pressure; URR, urea reduction ratio.

**Table 2 biomedicines-11-01026-t002:** Main determinants of left ventricular mass index in end-stage kidney disease, by a multiple regression analysis.

	End-Stage Kidney Disease Patients (*n* = 196)				
	Unstandardized Coefficients		Standardized Coefficients	t	*p* Value
Biomarkers	B	Standard error	β		
Leptin	−0.160	0.041	−0.263	−3.860	<0.001
NT-proBNP	0.240	0.060	0.273	4.012	<0.001

Note: data presenting a non-Gaussian distribution were transformed to data with normal distribution using the Templeton method [22]. NT-proBNP, N-terminal pro B-type natriuretic peptide.

## Data Availability

The data presented in this study are available on request from the corresponding authors. The data are not publicly available due to privacy and ethical restrictions.

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
