# Peer review of "The Association of Leptin with Left Ventricular Hypertrophy in End-Stage Kidney Disease Patients on Dialysis"

_biomedicines, 2023, doi:10.3390/biomedicines11041026_

Round 1

Reviewer 1 Report

The current study by Susana Coimbra et al. aimed to delineate the association of LVH and adipokine levels with NT-proBNP, GDF-15 along with BMI and albumin in ESKD patients on dialysis. The study has moderate patient group and has some concerns mentioned below.

11)      The authors should cite some papers either positive or negative about the role of leptin in CVD in the introduction section.

22)      The authors should also include graphs for all the significant data (BMI, NT-proBNP etc.) discussed on page 5 similar to leptin graph for a better visual representation.

33)      They should also include their study drawbacks in the discussion section.

44)      Proof read for minor grammar and spell check

Author Response

We would like to thank the reviewer for its positive appreciation of our work, and we thank the opportunity to further improve the manuscript with the pertinent comments/suggestions made.

The authors should cite some papers either positive or negative about the role of leptin in CVD in the introduction section.

R: As suggested, we included new information at Introduction: “Hyperleptinemia has been associated with inflammation, insulin resistance, protein energy wasting and with progression of CKD [14]. The increased leptin levels reported in CKD patients [15] were pointed as promoters of endothelial dysfunction, which is crucial for atherosclerosis development [16]. In peritoneal dialysis patients, leptin levels above 40 ng/mL were associated with LVH [17]. However, a study in ESKD patients under hemodialysis showed low leptin values as an independent predictor of mortality [18]. Decreased leptin levels were also associated with a higher risk of CV events in hemodialysis patients and was also considered an independent risk factor for LVH development [19]. Interestingly, in ESKD, while patients with a history of stroke presented higher leptin levels than those without stroke history, patients with congestive heart failure showed lower leptin than those without history of congestive heart failure [20]. Given the controversial data, further studies are warranted to fully clarify leptin role, namely, its association with morbidity and mortality risk and its contribution to LVH development in CKD and, especially, in ESKD.

The authors should also include graphs for all the significant data (BMI, NT-proBNP etc.) discussed on page 5 similar to leptin graph for a better visual representation.

R: We only observed significant differences between males/females with and without LVH for hemoglobin and NT-proBNP; no significant differences were observed between these subgroups for calcium and GDF-15. These results and a figure/graph for hemoglobin and NT-proBNP concentrations (like the graph used for leptin) were added to the manuscript, at Results section.

As depicted in Figure 2, ESKD males with LVH (11.3 [10.7-12.1] g/dL) presented significantly lower hemoglobin values (P = 0.029) than ESKD males without LVH (12.0 [11.3-12.7] g/dL); no significant differences (P = 0.515) were found between ESKD females with (11.3 [10.5-11.8] g/dL) or without LVH (11.4 [10.9-11.9] g/dL).

As observed in Figure 3, ESKD males with LVH (17.9 [10.0-30.9] g/dL) presented significantly higher NT-proBNP values (P = 0.003) than ESKD males without LVH (10.9 [5.7-17.3] g/dL); no significant differences (P = 0.188) were found between ESKD females with (13.6 [8.0-25.8] g/dL) or without LVH (10.2 [7.2-23.4] g/dL). For calcium and GDF-15 levels, no significant differences were observed between females with or without LVH or between males with or without LVH.”

They should also include their study drawbacks in the discussion section.

R: As suggested by the reviewer, we added the study drawbacks at the end of Discussion section: “The complexity of comorbid conditions and multidrug therapy (e.g. statins, diuretics, antihypertensives, erythropoiesis stimulating agents, iron) represents a limitation when interpreting data from ESKD patients; studies using larger samples would certainly reinforce the robustness of data, as well the performance of longitudinal studies. Nonetheless, in the present work, we performed a multiple regression analysis, allowing to study the contribution of different variables to LVMI values; we also performed a binary logistic regression analysis, allowing to determine the impact of each studied parameter on the likelihood of developing LVH, contributing to establish more accurate associations of LVH with the studied biomarkers and to reinforce conclusions.”.

Proof read for minor grammar and spell check.

R: A revision for grammar and spell check was made.

Reviewer 2 Report

Comments:

1. Could the authors please provide the ethical approval for their study please?

2. Could the the authors please clarify how the number of study participants were determined i.e. how was their study powered please?

Author Response

We would like to thank the reviewer for its positive appreciation of our work, and we thank the opportunity to further improve the manuscript with the pertinent comments made.

Could the authors please provide the ethical approval for their study please?

R: The reference to the ethical approval for this study was already presented in the submitted manuscript, at the “Institutional Review Board Statement”, in the end of manuscript, before References; however, we agree that it is a pertinent issue that should be in the text manuscript; thus, we add this information at “Material and Methods” section.

 “This study was conducted in collaboration with 5 Dialysis Clinics from the North of Portugal (from 2016-2019), after approval by the Committee on Ethics from Faculty of Pharmacy, Porto University, and by the Directors of Dialysis Clinics (Report No. 26-04-2016).”

Could the authors please clarify how the number of study participants were determined i.e. how was their study powered please.

R: The study presented in this article is part of a broader project (NephroCardioRisk), which included more than 300 patients on dialysis; only 196 of these were available for echocardiographic study. There are few studies on the prevalence of LVH in ESKD patients, referring to values between 75 and 90% (doi: 10.1681/ASN.V1251079; doi: 10.1038/ki.1995.22.; doi: 10.3109/08860220903100705; https://doi.org/10.1159/000435838).

Assuming that the prevalence of LVH is estimated to be close to 90%, the minimum sample size (using the Cochran`s formula) would be 98 for a confidence level of results of 90%, with a margin error of 5%; assuming that the prevalence of LVH in dialysis patients is 75%, the minimum sample size would be 204 for a confidence level of 90%, with a margin of error of 5%. The number of patients that we studied, 196, seems, therefore, sufficient to obtain reliable results.

Round 2

Reviewer 2 Report

The authors have satisfactorily responded to my comments.